# Signatures of Transcription Factor Evolution and the Secondary Gain of Red Algae Complexity

**DOI:** 10.3390/genes12071055

**Published:** 2021-07-09

**Authors:** Romy Petroll, Mona Schreiber, Hermann Finke, J. Mark Cock, Sven B. Gould, Stefan A. Rensing

**Affiliations:** 1Plant Cell Biology, Department of Biology, University of Marburg, 35037 Marburg, Germany; Romy.Petroll@bioinfsys.uni-giessen.de (R.P.); mona.schreiber@biologie.uni-marburg.de (M.S.); hermann.finke@biologie.uni-marburg.de (H.F.); gould@hhu.de (S.B.G.); 2Leibniz Institute of Plant Genetics and Crop Plant Research (IPK), 06466 Gatersleben, Germany; 3Algal Genetics Group, UMR 8227, Integrative Biology of Marine Models, Station Biologique de Roscoff, Sorbonne Université, CNRS, UPMC University Paris 06, CS 90074, 29688 Roscoff, France; cock@sb-roscoff.fr; 4Institute for Molecular Evolution, Heinrich-Heine-University Düsseldorf, 40225 Düsseldorf, Germany; 5Centre for Biological Signaling Studies (BIOSS), University of Freiburg, 79108 Freiburg, Germany

**Keywords:** Rhodophyta, transcription factor, morphological complexity, evolution, multicellularity

## Abstract

Red algae (Rhodophyta) belong to the superphylum Archaeplastida, and are a species-rich group exhibiting diverse morphologies. Theory has it that the unicellular red algal ancestor went through a phase of genome contraction caused by adaptation to extreme environments. More recently, the classes Porphyridiophyceae, Bangiophyceae, and Florideophyceae experienced genome expansions, coinciding with an increase in morphological complexity. Transcription-associated proteins (TAPs) regulate transcription, show lineage-specific patterns, and are related to organismal complexity. To better understand red algal TAP complexity and evolution, we investigated the TAP family complement of uni- and multi-cellular red algae. We found that the TAP family complement correlates with gain of morphological complexity in the multicellular Bangiophyceae and Florideophyceae, and that abundance of the C2H2 zinc finger transcription factor family may be associated with the acquisition of morphological complexity. An expansion of heat shock transcription factors (HSF) occurred within the unicellular Cyanidiales, potentially as an adaption to extreme environmental conditions.

## 1. Introduction

Oxygenic photosynthesis emerged about 2.4 billion years ago in the Cyanobacteria, and thus formed the basis for global formation of energy-rich substrates and oxygen [1,2]. With the increased concentration of oxygen in the atmosphere, the evolution of aerobic respiration, specific biosynthetic pathways, and multicellularity followed [3]. The inheritable plastid, that enables photosynthesis in eukaryotic cells, originates from endosymbiosis [4]. Primary endosymbiosis refers to the engulfment and subsequent modification of a free-living cyanobacterium-like bacterium by a heterotrophic eukaryote host [1]. The evolution of the first complex plastid through primary endosymbiosis occurred during the early emergence of the superphylum Archaeplastida, comprising red algae (Rhodophyta), Glaucophyta, and the “green lineage” (Chloroplastida)—the latter comprising the Chlorophyta and the Streptophyta, which can be further divided into streptophyte algae and land plants (Embryophyta) (Figure 1) [5]. The uptake of an already photoautotrophic eukaryote by another heterotrophic eukaryote is referred to as secondary or tertiary endosymbiosis, and the plastids are characterized by more than two surrounding membranes [4]. For example, red algae provided plastids to several other eukaryotic lineages, including diatoms, brown algae, haptophytes, cryptophytes, and dinoflagellates, via secondary endosymbiosis events [6,7].

Rhodophytes, a monophyletic group within the Archaeplastida, are characterized by the occurrence of phycobiliproteins, unstacked thylakoids, and the absence of flagella and centrioles [7]. The red algae are a species-rich group that include unicellular as well as large, multicellular taxa. Red macroalgae are one of five eukaryotic groups (animals, Chloroplastida, fungi, red algae, and brown algae) that independently developed complex multicellularity [6]. Furthermore, the oldest known fossils of complex multicellular organisms have been assigned to the red algae lineage [8]. In addition, the findings of Bengtson et al., which include the discovery of a part of the photosynthetic machinery of red algae in about 1.6-billion-year-old fossils, led to the updated appearance of an early red algae by about 400 million years earlier than previously thought [9]. Red algae can be found in many marine ecosystems and are ecologically important in the intertidal and subtidal flora on rocky shores [6,7]. The Rhodophyta can be classified into two subphyla, Rhodophytina and Cyanidiophytina, and are subdivided into seven classes, namely Florideophyceae, Bangiophyceae, Cyanidiophyceae, Compsopogonophyceae, Porphyridiophyceae, Rhodellophyceae, and Stylonematophyceae [10]. 

Previous studies have suggested that the compact genome structure and the low number of introns in most red algal genes are the result of an “evolutionary bottleneck” during the emergence of this lineage [6,7]. This hypothesis proposes that the unicellular ancestor of red algae went through contraction of genome size due to selection pressure caused by extreme environmental conditions (Figure 1) [7]. Accordingly, a major driver for the apparent gene loss was adaption to life in an extreme environment [11]. For this study, we will refer to the genome contraction early during the evolution of the red lineage as “genome contraction event” and will use the term “secondary” (for example, for describing evolutionary subsequent expansion events) with regard to this frame of reference.

Members of the order Cyanidiales are extremophilic, asexual, unicellular organisms that live under extreme temperature and acidic conditions, and they are considered to be most similar among the extant red algae to the putatively thermo-acidophilic unicellular most recent common ancestor (MRCA) of the red algae [6,12]. Ciniglia et al. proposed that the habitats of the extant Cyanidiales are exclusively thermal and acidic sites, but with a broad range of temperatures and pH values observed, varying from 18 to 55 °C, and pH 0.5 to 2 [12]. With respect to the genome contraction hypothesis, the Cyanidiales might therefore still be under the adaptive pressure that shaped the MRCA, while non-extremophile members of the Porphyridiophyceae, Bangiophyceae, and Florideophyceae would have subsequently undergone genome expansions [7]. Gene duplications with subsequent paralog retention are thought to be one of the main drivers of morphological complexity, e.g., in land plants [13]. However, paranome-based analyses of the multicellular red alga *Chondrus crispus,* a member of the Florideophyceae, yielded no evidence for whole-genome duplications [7]. It did, however, reveal that three segmental or chromosomal events followed by paralog retention had taken place in the *C. crispus* lineage. Lee et al. [14] investigated possible lineage-specific gene duplication events in *Gracilariopsis chorda* (also a member of the Florideophyceae) using k-mer analysis, and concluded that no large-scale gene duplications or polyploidization events had occurred either. Still, they found that genome expansion had occurred in the Florideophyceae through the accumulation of transposable elements [14]. 

A frequently studied issue, in various research areas, is a unified definition for multicellularity, and in particular for complex multicellularity [5]. So far, no definition has been found that unifies the actual principle behind cellular complexity across all organisms or systems [5]. However, if analyses based on multicellularity are to be conducted, proxies provide an option. One suggested proxy is the genome size of an organism, but genome size does not correlate with the organismal complexity, an observation known as the c-value enigma [5,11]. Another proxy that can be used is the number of cell types [6]. This is only an approximation, however, since the exact number of different cell types can usually not be determined. For some analyses, a specific number of cell types can be defined, above which a species is defined as complex multicellular [6]. Within the red algae, there are different levels of cellular complexity, varying from simple filaments to more complex forms in the Florideophyceae [6]. The Cyanidiales and Porphyridiophyceae include only unicellular species, whereas the Florideophyceae and Bangiophyceae include taxa with advanced multicellularity (using the number of cell types as a proxy for complex multicellularity) [6,10,15]. The Florideophyceae are exclusively multicellular [6]. A frequently studied question is how and why multicellularity emerged, and in particular, whether there is a specific class of genes that shows an association with the occurrence of multicellularity and could therefore serve as the aforementioned proxy [5].

In addition to genome size and the number of cell types, the complexity of gene regulatory networks is another suggested proxy. Gene regulatory networks (GRNs) play a role in the expression of various functions, in the development as well as morphology of both unicellular and multicellular organisms [16]. Here, transcription-associated proteins (TAPs) play a critical role. TAPs are important components of developmental and signaling processes [17]. Since TAPs are widely distributed but also show lineage-specific patterns, they are well-suited for performing phylogenetic analyses [16]. They can be divided into transcription factors (TFs), transcriptional regulators (TRs), and putative TAPs (PTs). A TF binds to a specific sequence of cis-regulatory elements and enhances or represses transcription [16,18]. A TR can have different functions, it can be part of the transcriptional core complex, may be involved in protein–protein interactions, or can be responsible for chromatin modifications [17,18]. Lang et al. performed a phylogenetic comparative (PC) analysis to investigate the correlation between TAPs and morphological complexity (measured by the number of cell types) in the Chloroplastida, and found that the complement of TFs encoded by a genome can be used as a proxy for multicellularity in that lineage. Based on this PC analysis, it was hypothesized that, as in multicellular animals, the expansion of GRNs and TAPs was a major factor in the evolution of morphological complexity in the Chloroplastida [5]. In this context, a TF family that has been highlighted in various studies is C2H2. Albertin et al. [19] investigated the relationship between new morphological features, including features of the complex nervous system, and the expansion of certain gene families in the octopus species *Octopus bimaculoides* compared to other invertebrate bilaterians. It was concluded that expansion of the C2H2 gene family in octopus correlates with the acquisition of morphological complexity. The same trend was observed for *C. crispus*. Based on a cross-species partial least squares (PLS) analysis, Collén et al. found that the C2H2 family is twice as abundant in *C. crispus* as in *Cyanidioschyzon merolae,* a member of the Cyanidiales [7]. Some exceptions to the hypothesis of a correlation between TAP family size and multicellularity, however, have been recently identified [20,21,22]. Expansions of specific TAP families in response to certain environmental conditions and ecological niches have been suggested as a possible reason [20], as well as secondary contraction [23].

Here, we present an analysis of algal TAP families in order to screen for significant differences in the occurrence of specific families in unicellular versus multicellular members of the Rhodophyta, and thus whether TAP complements are correlated with the emergence of multicellularity in the red algae. Based on TAPscan [17], we updated selected profile hidden Markov models (HMMs) of six domains and added two new domains, with a special focus on improving the sensitivity to detect TAPs in algae. This updated TAPscan v3 was used to analyze ten red algae species to investigate the evolution of morphological complexity and the occurrence of TAPs in red algae.

**Figure 1 genes-12-01055-f001:**
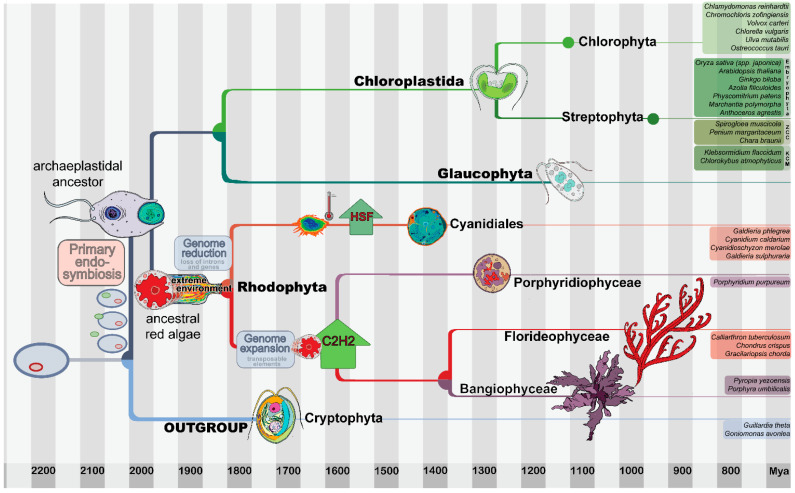
The evolution of the Archaeplastida with a focus on red algae. The Archaeplastida are characterized by the synapomorphic gain of a primary plastid (surrounded by two membranes). Archaeplastida comprises the Chloroplastida (green lineage, including land plants within the Streptophyta), the Glaucophyta, and the Rhodophyta (red algae). For the evolutionary analyses conducted in this study, the archaeplastidal tree of life was outgroup-rooted using the Cryptophyta, which are a sister group [24]. The most recent common ancestor (MRCA) of red algae is hypothesized to have gone through a phase of genome contraction caused by adaptation to extreme environments. The extant, exclusively unicellular Cyanidiales live in extreme environments and are a sister group to the Porphyridiophyceae, Bangiophyceae, and Florideophyceae. Due to the multicellular members of the Bangiophyceae and the entirely multicellular Florideophyceae, secondary gains of complexity (after the genome reduction event) are evident in both latter lineages. The tree as well as the timeline are based on a recently published molecular clock analysis [2] and comprehensive phylogenetic analyses [25,26]. The nodes within the phylogeny are placed according to the data split times (see bottom timeline), and the position of the cartoons of the analyzed species represent the first split within the respective lineage. Primary endosymbiosis, genome reduction and expansion, as well as the HSF expansion and C2H2 abundance discussed in this work are depicted. The species used for the analyses are listed to the right.

## 2. Materials and Methods

### 2.1. Dataset 

The Rhodophyta dataset used in this analysis consisted of ten red algal species, including three members of the Florideophyceae, two members of the Bangiophyceae, one member of the Porphyridiophyceae, and four members of the Cyanidiales (Figure 1). A set of thirty species was used in the phylogenetic analysis, consisting of the above ten members of the Rhodophyta, plus two members of the Cryptophyta, six members of the Chlorophyta, two streptophyte algae of the KCM grade (comprising the basal lineages Klebsormidiales, Chlorokybales, and Mesostigmatales), three streptophyte algae of the ZCC grade (comprising the sister lineages of extant land plants, namely Zygnematophyceae, Coleochaetophyceae, and Charophyceae), and seven members of the Embryophyta. The names, five-letter codes, and the data sources of the species used are shown in Appendix A.

### 2.2. Improving the Sensitivity of Selected Profile HMMs for Algae by Adding Diverse Algal Sequences 

In previous studies, TAPscan was introduced as a tool to detect TAP families based on protein domains using profile HMMs and specific classification rules [16]. A TAPscan update together with the presentation of the TAPscan web interface followed [17]. Here, we used TAPscan to detect 122 TAP families and to annotate the appropriate protein sets. The steps of updating TAPscan, which are described in the following, are schematically shown in Figure 2. Previous TAPscan analyses of the brown alga *Ectocarpus siliculosus* and the above-mentioned update of TAPscan described by Wilhelmsson et al. [17] indicated that it would be beneficial to increase the sensitivity of some specific profile HMMs for their use with algae. These previous analyses were largely based on manual annotations of the *E. siliculosus* TAPs, using the TAPscan v2 output as a starting point (Appendix A). The additional manual annotations included a keyword search in ORCAE [27], various BLAST searches [28], and phylogenetic analyses to group specific matches into subgroups and to eliminate false positives. Special focus was given to a few specific domains/families of interest. These domains were zf-C2H2, AP2, bZIP_1, bZIP_2, HLH, and HMG_box, corresponding to the TAPs C2H2, AP2/EREBP, bZIP, bHLH, and HMG. These families were selected because, based on the manual annotation, it was determined that the sensitivity of TAPscan could be improved for *E. siliculosus*, i.e., not all manually curated gene family members were detected by TAPscan v2. To increase sensitivity, the first step was to update the 118 HMM profiles recovered from PFAM version 29.0 (December 2015) to a more recent version (33.1, May 2020) by downloading the profiles from the PFAM website [29]. For the domains Auxin_resp, bZIP_2, and JmjC, the gathering cutoffs from PFAM version 29.0 were used, and for K-box, the gathering cutoff established previously (TAPscan v2) was used [17]. More of the expected sequences of these families were recovered using the earlier, lower cutoffs than using the PFAM version 33.1 cutoffs, so these cutoffs were replaced with these earlier parameters. 

The next step was to add additional sequences from red algae, brown algae, SAR-group (Stramenopiles, Alveolates, and Rhizaria) members, and streptophyte algae to the seed alignments (downloaded from PFAM) underlying the profile HMMs. This approach has previously been successfully used to update TAPscan from v1 to v2 and is based on the fact that most PFAM domain seed alignments are taxonomically biased towards (usually) animal and seed plant model organisms. In order to add sequences to the alignments, appropriate sequences were first collected and stored as fasta files. To collect the sequences, TAPscan runs were performed for all species belonging to the above-mentioned groups. The names of sequences were saved in text files; subsequently, seqtk version 1.3 [30] was used to retrieve the sequences according to the sequence names from the corresponding protein sets, and the sequences were stored as fasta files. Which of these sequences were selected for incorporation was decided based on two criteria: First, the completeness of the sequences was verified by manual assessment. The second criterion was that, if possible, sequences from all the groups were used (brown algae, red algae, SAR-group members, streptophyte algae) and from as many different species as possible. In addition, the new sequences were not allowed to exceed 25% of the total number of sequences of the seed alignment, in order not to introduce too many novel sequences into the existing seed alignment. If more than the 25% of the total number of sequences fulfilled the criteria, new sequences were randomly eliminated. MAFFT version 7 was used to construct an alignment for each of the six domains [31]. The parameter --add was used to add unaligned sequences into the existing alignment. In addition, --auto was used to automatically optimize the algorithm according to the data size and properties. The alignment was then trimmed using BioEdit version 7.2.5 [32] and manually curated so that only the domain of interest was present in the alignment. The alignment was trimmed based on the original seed alignment from PFAM, and the sequences were rechecked for completeness. New profile HMMs were calculated from these alignments using hmmbuild from the HMMER software package version 3.3.2 [33]. The alignments and profile HMMs are available in Appendix A.

### 2.3. New TAPscan Version

Even after improving the sensitivity of the profiles for the domains of particular interest, not all the expected bZIP *E. siliculosus* sequences were detected. To further investigate these missing sequences, a search was performed using InterPro 83.0 [34]. For the TAP bZIP, 23 hits were expected based on the manual annotation, but only 21 were detected using the updated profiles. Since four of the detected sequences differed from the expected sequences, an InterPro search was carried out for the bZIP domains. This analysis detected a conserved domains database (CDD) entry “bZIP_AUREO-like” (cd14809) and an unintegrated bZIP, CDD entry (cd14686). Again, alignments of these domains were downloaded and profile HMMs were calculated (bZIP_AUREO and bZIP_CDD). These profile HMMs were added to better define the bZIP family. Thus, the rules “bZIPAUREO; bZIP_AUREO; should”, “bZIPAUREO; HLH; should not”, and “bZIPAUREO; Homeobox; should not”, and the same rules equivalent to bZIPCDD were added to the TAPscan v3 rule set. In addition, the appropriate changes to the TAPscan script and determined coverage values were added. Coverage values of 0.75 and 0.50 were used for bZIP_AUREO and bZIP_CDD, respectively. Using this approach, 26 bZIP hits were detected in *E. siliculosus.* Of the five newly detected sequences, three sequences were expected sequences and two were additional hits. Based on this result, we decided to use these two HMM profiles in addition to bZIP_1 and bZIP_2. 

After improving the HMG_box HMM profile, it was noticed that four of the expected sequences based on manual annotation were assigned to families PHD, CCAAT_HAP5, and two sequences to SWI/SNF_SNF2 instead to HMG_box. Since the non-HMG profiles provided lower (i.e., stronger) E-values, the sequences were assigned to the non-HMG domains. However, based on the manual annotation, it was observed that the four missing proteins were multi-domain proteins and thus could also have been assigned to HMG_box. Therefore, we added the rule “HMG_box; should not” to the rules for PHD, CCAAT_HAP5, and SWI/SNF_SNF2. All 14 expected sequences were thus assigned to the domain HMG_box in *E. siliculosus*.

Furthermore, it was assessed whether the gathering thresholds (GA-threshold), representing the suggested minimum score a sequence must provide to be assigned to a domain, should be adjusted for the improved domains. For that, we used recent studies of *Arabidopsis thaliana* (when available) and calculated sensitivity and specificity (Appendix A). For zf-C2H2, we used the analysis of Englbrecht et al. [35]. Based on this comparison, we decided to use the GA-threshold of 9.45 employed by Wilhelmsson et al. [17]. For bZIP, we used the original PFAM threshold based on the comparison with the analysis by Dröge-Laser et al. [36]. To investigate bHLH, we used the study of Zhang et al. [37]. For this profile, we decided to use a GA threshold of 14.00. The integration of the above-mentioned profile, script, and TAPscan rule changes resulted in a new version of TAPscan, v3. 

### 2.4. Comparisons of Unicellular and Multicellular Red Algae 

To compare TAP family sizes in ten red algal genomes, averages were calculated and compared using Microsoft Excel. Statistical tests (T-tests or Wilcoxon rank sum tests, depending on whether a Shapiro–Wilk normality test classified the data as normally distributed or not) were then performed using R version 3.6.3 [38], and the results are shown in Appendix A. A principal component analysis (PCA) was carried out using R and visualized using the ggplot2 [39] and ggbiplot packages [40]. For this analysis, 62 different TAP families were used, for which at least one gene occurs in one of the red algae.

### 2.5. Calculation of Expansions/Contractions and Gains/Losses 

The count package [41] was applied to investigate expansions/contractions and gains/losses in the Rhodophyta (Appendix A). In particular, asymmetric Wagner parsimony was used, with default settings. For the species phylogeny, a modified version of the species tree (astral-33-new-renamed.tre, based on 410 single-copy genes) reported by Leebens-Mack et al. [26] was used, containing only the species used in this study. The clustering of *Pyropia yezoensis* was adapted so that the two Bangiophyceae form a clade. In addition, the clustering of *Porphyridium purpureum* was modified according to the phylogeny of Muñoz-Gómez et al. [25], resulting in the Porphyridiophyceae representing a sister lineage to the Florideophyceae and Bangiophyceae clade [25]. Since *C. merolae* was the only species in the tree for the Cyanidiales, the branch lengths for *Galdieria phlegrea* and *Galdieria sulphuraria* were taken from Qiu et al. [42] and were adjusted and set in relation to the branch length from the species tree reported by Leebens-Mack et al. [26]. The branch length for *Cyanidium caldarium* was set based on the species distribution in the phylogenetic tree reported by Miyagishima et al. [43]. Our dataset contains the TAPs for 30 species belonging to the Cryptophyta, Rhodophyta, Chlorophyta, Streptophyta, and Embryophyta. The Cryptophyta were used as an outgroup in the species tree. 

### 2.6. Phylogenetic Inference

Phylogenetic trees were inferred for the TAP families C2H2 and HSF (Appendix A). First, to recover the corresponding (orthologous) sequences from the proteins of the 30 species from the Rhodophyta, Cryptophyta, Chlorophyta, streptophyte algae, and Embryophyta, the -A option of hmmsearch from the HMMER package [33] was used to save the protein sequences of all detected TAPs. The output was then filtered for C2H2 and HSF sequences. MAFFT version 7 with the --auto option was used to construct multiple sequence alignments for C2H2 and HSF. All gaps between the conserved regions were deleted in the HSF alignment using BioEdit [32]. Jalview version 2.11.1.3 was used to visualize the alignments [44]. Quicktree version 2.0 was used to calculate neighbor joining trees to provide initial estimates of the phylogenetic trees [45]. Maximum likelihood trees were constructed using IQ-TREE multicore version 1.6.1 with the MFP (ModelFinder Plus) option to find the best evolutionary models [46]. For HSF, the model LG + R5 was used, for C2H2 VT + R8. A Bayesian inference tree was constructed using MrBayes version 3.2.5 for the HSF family [47]. The best evolutionary model was calculated using ProtTest version 3.4.2 [48]. For HSF, the model LG was used. FigTree version 1.4.4 was used to visualize and color the trees [49]. 

## 3. Results and Discussion

### 3.1. Improved Sensitivity for the Detection of Five TAP Families

TAPscan [17] was used to analyze the occurrence of TAPs in red algae. We previously demonstrated that a more taxonomically balanced profile HMM is able to perform with higher sensitivity [17]. During analyses and manual curation of algal genomes, we found that several TAP family profiles had suboptimal sensitivity, in particular AP2/EREBP, bZIP, bHLH, C2H2, and HMG (Appendix A). Hence, we decided to add sequences from red algae, brown algae, members of the SAR-group, and streptophyte algae to the seed alignments of these six profile HMMs (representing the five above-mentioned TAP families) to overcome their taxonomic bias and to increase their sensitivity for algae. Furthermore, two new profile HMMs for better definition of the bZIP family were added (bZIP_AUREO and bZIP_CDD), corresponding rules were added to the rule set, and some GA-thresholds were adjusted (cf. Materials and Methods, Section 2.3). The TAPscan version v3 reported here exhibits increased sensitivity for all five of the above TAP families, with a minimum of two and a maximum of twelve additional true positives detected. As a gold standard, we used a manual curation of these families for the brown alga *E. siliculosus* and found that sensitivity for all five domains had increased (Appendix A). We also compared C2H2 and bHLH to recent studies that had annotated these families [35,37] and calculated sensitivity and specificity using them as gold standards (Appendix A). We assessed whether TAPscan performance was altered for seed plants and found that sensitivity was either the same or improved, and that specificity was unaltered. The adjusted alignments and profile HMMs are available in Appendix A. The new TAPscan v3 was used to perform annotation of TAPs for a set of ten unicellular and multicellular red algae (Appendix A). These species have been added to the TAPscan web interface and can be accessed there (https://plantcode.online.uni-marburg.de/tapscan/ (accessed on 1 May 2021). 

### 3.2. Differential Occurrence of TAP Families in Unicellular and Multicellular Red Algae

To investigate whether (and which) TAP families expanded concomitantly with the emergence of multicellularity in the red algae (Figure 1), we analyzed the occurrences and numbers of TAPs in four species of the order Cyanidiales, three species of Florideophyceae, two multicellular species of the Bangiophyceae, and one member of the Porphyridiophyceae, and visualized the results using PCA (Figure 3). The first PC separates the unicellular Cyanidiales from the Porphyridiophyceae, Bangiophyceae, and Florideophyceae. The Cyanidiales inhabit extreme environments, i.e., hot and acidic habitats, potentially causing adaptive pressure to these environments (Figure 1) [7,42]. The Porphyridiophyceae, Bangiophyceae, and Florideophyceae are thought to have undergone a secondary expansion of their genomes [7]. Indeed, we find that their TAP gene repertoire is more complex as compared to the Cyanidiales, that are well-separated in the PCA (Figure 3) [7]. Interestingly, the unicellular Porphyridiophyceae species *P. purpureum* (PORPU) clusters with the multicellular Bangiophyceae and not with the unicellular Cyanidiales. This suggests that genome expansion may already have occurred in the unicellular common ancestor of Porphyridiophyceae, Bangiophyceae, and Florideophyceae, after separation of the Cyanidiales, leading to increased diversity of TAPs.

### 3.3. The Sizes of the C2H2 and HSF Families Differ Significantly across the Red Algae

We performed statistical testing to further investigate TAP family evolution (Appendix A). In the green lineage, a correlation has been observed between the total number of TAPs and morphological complexity [16]. No significant difference, however, was detected when the total numbers of TAPs were compared between uni- and multi-cellular red algae (*p* = 0.26, *t*-test). Differences for individual TF families, particularly for the heat shock factor (HSF) and C2H2 zinc finger families, were detected (Figure 4). The C2H2 family has been hypothesized to be associated with a shift towards multicellularity (e.g., [7,19]). Individual studies of C2H2, for example in *Medicago truncatula,* show that they may act as a determinacy factor, regulating spatial-temporal expression of other genes and hence impacting leaf morphogenesis [50]. C2H2 is one of the largest TF families in plants, and its defining domain enables proteins to bind to nucleic acids and proteins [51]. There were significantly (*p* = 0.03, Wilcoxon rank sum test) more C2H2 members in multicellular than in unicellular red algae, a trend also observed from plotting the numbers (Figure 4). This finding was consistent with the previous analysis of the first multicellular red algal genome, *C. crispus* (as compared to the unicellular species *C. merolae*) [7]. Moreover, increased C2H2 abundance also coincides with morphological complexity in the invertebrate octopus [19]. In mammals, abundance of C2H2 correlates with species-specific morphogenetic patterns and functions [52], and C2H2 expansions in eukaryotes have been shown to be associated with morphological complexity [53]. A possible correlation between C2H2 expansion and the emergence of multicellularity in red algae could be explained by two evolutionary scenarios: the gain of C2H2 genes after the divergence of Cyanidiales and the other lineages, or the loss of such genes in Cyanidiales. The pattern of the domain trees (Appendix A) supports the former scenario, independent gain after the divergence from the Cyanidiales: the majority of sequences from the Porphyridiophyceae/Bangiophyceae/Florideophyceae form clades that do not contain sequences from any other species. These genes very probably were gained and retained within this monophyletic group. Moreover, the low numbers of C2H2 genes in Chlorophyceae (up to 14) also suggest that the MRCA of Archaeplastida did not harbor many C2H2 genes, rendering the secondary loss scenario less likely. Notably, regardless of a loss or gain scenario, high C2H2 abundance was also observed for the unicellular Porphyridiophyceae *P. purpureum.* This might suggest that an abundance of the gene family in the MRCA of the Porphyridiophyceae/Bangiophyceae/Florideophyceae laid the ground for the emergence of multicellularity in part of this clade. However, more data of unicellular Porphyridiophyceae/Bangiophyceae are needed to infer further conclusions.

HSFs are known to be involved in reactions to extreme environmental conditions and chemical and physiological stress [54]. The numbers of HSFs in unicellular and multicellular red algae were not significantly different (*p* = 0.06, Wilcoxon rank sum test). Between the Cyanidiales vs. multicellular members of the Bangiophyceae and Florideophyceae, however, it was significant (*p* = 0.04, Wilcoxon rank sum test), indicating that significantly more HSFs are present in unicellular Cyanidiales as compared to multicellular red algae (Figure 4). Extant Cyanidiales live under extreme conditions, and their gene set is presumably adapted to these conditions. Collén et al. hypothesized that there was less competition in extreme environments at the time of the unicellular MRCA of the red algae because only a few cyanobacterial lineages were adapted to these acidic conditions at that time [7]. In extant Cyanidiales, a wider range of temperatures and pH values can be observed. To adapt to varying temperatures up to 55 °C likely required a more diverse HSF family to enable a flexible heat shock response [42,54]. Furthermore, the presence of just a single HSF in the unicellular Porphyridiophyceae *P. purpureum* is more similar to the number in the multicellular Bangiophyceae (all have either one or zero HSFs) than to the distributions in the Cyanidiales (one to six HSFs). 

### 3.4. Evolution of the TAP Families C2H2 and HSF 

To further analyze the C2H2 and HSF families, we added a phylogenetically diverse group of 20 species comprising Cryptophyta, Chlorophyta (Chlorophyceae, Trebouxiophyceae, Ulvophyceae, Prasinophyta), streptophyte algae (ZCC grade, KCM grade), and Embryophyta (monocots, eudicots, Acrogymnospermae, Bryophyta, Moniliformopses, Marchantiophyta, Anthocerotophyta) to the ten species of red algae and inferred protein family trees (Figure 5 and Appendix A). The Cryptophyta, a sister lineage to the Archaeplastida, were used as an outgroup [2,24]. In the maximum likelihood tree of the HSF family (Figure 5), the Cryptophyta clade was well-supported (SH-aLRT and ultrafast bootstrap support values of 100%). The red algae form a clade with high bootstrap support that is subdivided into two clades. Clade 1 features sequences of the Cyanidiales, the Porphyridiophyceae, Bangiophyceae, and the Florideophyceae. Clade 2 exclusively features sequences of the Cyanidiales. The higher number of HSF genes in Cyanidiales compared to members of the Porphyridiophyceae, Bangiophyceae, and Florideophyceae could be explained by two scenarios. First, the unicellular ancestor of the red algae may have possessed more than one HSF, and some members were lost in the Porphyridiophyceae, Bangiophyceae, and Florideophyceae after separation from the Cyanidiales lineage. Second, HSF expansions occurred in the Cyanidiales after their divergence from the common red algal ancestor. The tree (Figure 5) is in accordance with the latter scenario, suggesting that an ancestral HSF gene was present in the red algae MRCA, and that gene duplications with subsequent paralog retention occurred in the Cyanidiales.

Jiao et al. [20] analyzed the genomes of the unicellular streptophyte algae *Penium margaritaceum* and *Spirogloea muscicola* and compared them to that of the multicellular streptophyte alga *Chara braunii.* They detected fewer TAPs in the multicellular as compared to the unicellular species. Here, we detected eight HSFs in *S. muscicola* but only two in the multicellular *C. braunii.* Additionally, the total number of TAPs in *S. muscicola* (1168) was significantly higher than in *C. braunii* (699 TAPs). Jiao et al. concluded that less structural complexity is not always correlated with fewer TAPs and that expansions of specific TAP families can also occur through horizontal gene transfer (HGT) or segmental gene duplications [20]. Examining the tree, it is noticeable that three *S. muscicola* sequences, and four sequences in the collapsed groups, cluster together and are highly conserved with short branch lengths. This indicates lineage-specific expansion characterized by gene duplication events, as in the Cyanidiales. In general, it is observed that the distribution of the HSF family in red algae is not consistent with the trend that more TAP members are present in morphologically more complex organisms [16], since unicellular red algae have more HSF members than multicellular species. This is also the case for HSFs in the streptophyte algae *P. margaritaceum* and *S. muscicola.* In both algal groups, lineage-specific expansions seem to have occurred, potentially as adaptations to environmental conditions. 

We also inferred protein family trees for the TAP family C2H2 using the same group of species (Appendix A). The trees demonstrate that significantly more C2H2 genes occur in multicellular Bangiophyceae and Florideophyceae than in the Cyanidiales, and that most of the genes were gained and retained after the divergence from the Cyanidiales. This C2H2 abundance might be related to increased morphological complexity in the respective groups. As outlined above, the numbers in *P. purpureum* suggest that many genes might have been gained before the divergence of Porphyridiophyceae and Bangiophyceae/Florideophyceae, and hence the MRCA of the whole clade might have been equipped with the potential (a “C2H2 abundance playground”) for multicellularity to arise in some of the lineages. 

### 3.5. Changes in the Presence and Abundance of TAP Families during Evolution

We used a modified species tree (Figure 6) based on the species tree reported by Leebens-Mack et al. [26] to analyze gain, loss, expansion, and contraction of TAP families. Analysis using asymmetric Wagner parsimony implemented in the count tool [41] indicated a total of 93 expansions, 13 contractions, 32 losses, and 90 gains across all nodes (Figure 6, Appendix A). Of the 32 losses, 22 were detected in the red algae, while only 8 of the 93 expansions occurred in the same group. These statistics reflect the proposed massive genome reduction in the MRCA of rhodophytes. In contrast, most of the gains and expansions occurred during the evolution of the Phragmoplastophyta, in line with previous studies [17,20,22,55].

The occurrences of some TAP families in the red algae show unusual patterns. The TR Coactivator p15 is present in three of the four Cyanidiales and the Porphyridiophyceae *P. purpureum*, but not in any multicellular red algae. The presence of this family therefore coincides with unicellularity, which is interesting given that gene loss events may also play a role in enabling the transition to multicellularity [56]. In addition, loss of the zinc finger protein ZPR1, also a TR, was detected for all members of the Florideophyceae. Several gains were also detected: the TF ARF in *C. crispus*, the TR Argonaute in the Porphyridiophyceae/Bangiophyceae/Florideophyceae, and the TF C2C2 CO-like in two of the Florideophyceae species. Concerning the detected gain of the TR Argonaute in the Porphyridiophyceae/Bangiophyceae/Florideophyceae, which might imply potential HGT, its loss in the Cyanidiales would be an equally parsimonious interpretation. Additional expansions were detected of the Argonaute TR family in *C. crispus* and *P. purpureum*, and of the MADS TF family in the Bangiophyceae and *P. purpureum* (Figure 6; note that whether a gain or loss is predicted depends on sampling depth, and the most parsimonious interpretation may not necessarily correspond to the actual history due to sampling bias). Notably, ARF, C2C2 CO-like, MADS, and C2H2 have previously been described as putatively related to plant complexity evolution [5]. Some Porphyridiophyceae-, Florideophyceae-, and Bangiophyceae-specific expansions were observed, for example the organellar Sigma70-like TF. This might be related to polyplastidy, a trait associated with macroscopic, multicellular body plans among the Archaeplastida which evolved several times independently [57]. However, the monoplastidic *C. merolae* and *G. sulphuraria*, both Cyanidiales, also feature several Sigma70-like proteins, while the family was lost in the Florideophyceae *Calliarthron*
*tuberculosum*. Expansion of the HSF family in the Cyanidiales was confirmed by asymmetric Wagner parsimony, while the C2H2 family expansion in multicellular red algae or a contraction of this family in the Cyanidiales was not detected using this method. 

Qiu et al. [42] investigated the genome of *G. phlegrea,* a member of the Cyanidiales. By comparing the genome data with the data of other red algae, they showed that of the 6801 orthologous gene families that were assumed to be present in the ancestor of the red algae, 1448 were lost in the common ancestor of the Cyanidiales, but just 456 in the common ancestor of the Bangiophyceae and Florideophyceae. It was suggested that HGT could have played a role in regaining adaptive functions in the Cyanidiales [42], illustrated by the gain of a complete set of genes for the urea hydrolysis from Eubacteria in *G. phlegrea*. This could be an artifact, however, as the “last one out” always looks like a HGT [58], and for this set of genes, an extensive loss in the ancestor of the Cyanidiales was confirmed [42]. We also observed a divergence in the occurrence of different TAP families in the Cyanidiales. In particular, the expansion of the HSF family could be explained by different HGT events or (more likely) by lineage-specific duplication events, to compensate for gene loss after the genome contraction and to potentially allow the lineage to adapt to hot, acidic, and stressful environments. 

## 4. Conclusions

In this study, we aimed to determine whether there is a relationship between the overall size of the TAP complement (or the sizes of individual families) and the occurrence of multicellularity in the red algae. In the Chloroplastida, there is a correlation between the total TAP complement and morphological complexity, as measured by the number of different cell types [16]. In contrast, the red algae possess few TAPs overall, and this has been proposed to be the result of genome reduction as the result of passing through an “evolutionary bottleneck” ca. 1900 Ma ago [2]. Indeed, no significant differences could be detected between the total numbers of TAPs in unicellular and multicellular red algae. However, the TAP family complement of the unicellular Cyanidiales was found to deviate from the TAP family complement of the Florideophyceae and Bangiophyceae. This observed expansion of the TAP complement that likely occurred after the split from the Cyanidiales coincides with an increase in morphological complexity (as evidenced by the evolution of multicellular body plans). TAP complement expansion was also observed in the unicellular species *P. purpureum* belonging to the Porphyridiophyceae. Hence, the MRCA of the Porphyridiophyceae, Bangiophyceae, and Florideophyceae had already acquired/expanded certain TAP families that the Cyanidiales lack. 

A coincidence between the number of C2H2 TFs and organism complexity has been observed for the invertebrate octopus and the multicellular red alga *C. crispus* (compared to other invertebrate bilaterians and *C. merolae*, respectively), and it was argued that the expansions of these zinc finger TFs were related causally to increases in morphological complexity in both lineages [7,19]. Here, we show that multicellular and thus morphologically more complex red algae harbor significantly more C2H2 TF genes than their unicellular sister lineage Cyanidiales. Possibly, the size of the C2H2 family may correlate with the gain in morphological complexity similar to lineages as diverse as mammals, octopus, and multicellular red algae. It would be interesting to study more unicellular species belonging to the Porphyridiophyceae and Bangiophyceae, to check whether those (like the single unicellular *Porphyridium* species analyzed here) also encode more C2H2 genes than the Cyanidiales.

Analysis of the HSF TF family indicated that TAP family expansion did not occur exclusively in the Porphyridiophyceae, Bangiophyceae, and Florideophyceae, but also in the unicellular Cyanidiales. Probably, the MRCA of all red algae possessed an ancestral HSF gene, and while the Porphyridiophyceae, Bangiophyceae, and Florideophyceae retained that single gene (respectively lost it in the case of *C. tuberculosum*), HSF gene family expansion occurred in the Cyanidiales. The extremophilic Cyanidiales have presumably adapted their gene repertoire to be viable under environments with extreme temperatures or acidity. The HSF TFs are potentially involved in the regulation of the heat shock response. 

In conclusion, a general tendency of TAP complement correlation with multicellularity or complexity in the red algae is not observed, but the size of TAP families—particularly true for the C2H2 family—does coincide with it. Moreover, the pattern of lineage-specific expansion of TAP families may also reflect the environmental condition an organism is exposed to and reveal one mechanism as to how it may adapt.

## Figures and Tables

**Figure 2 genes-12-01055-f002:**
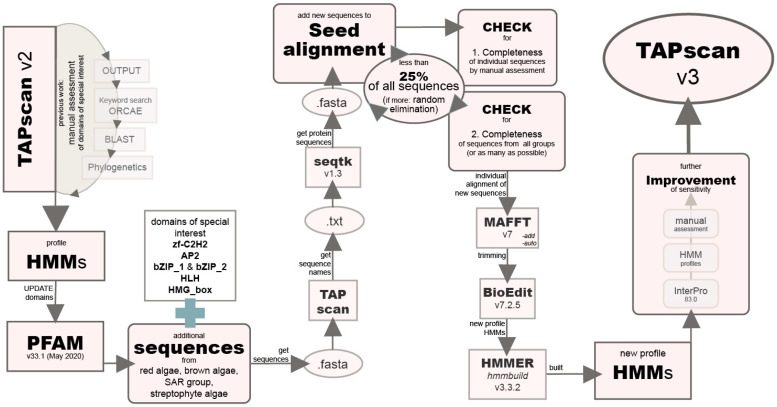
Flow chart illustrating the steps of improving the sensitivity of selected profile hidden Markov models (HMMs) towards algae by adding diverse algal sequences. The first step was to update all PFAM domains included in TAPscan v2 to the latest version (PFAM v33.1). Subsequently, additional sequences were added to the seed alignments of six domains of special interest and new profile HMMs were calculated. After further adjustments (described in Section 2.3), TAPscan is now v3.

**Figure 3 genes-12-01055-f003:**
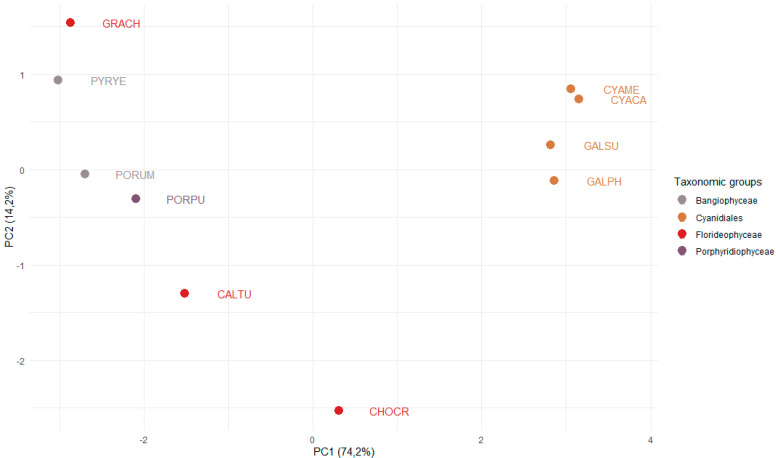
PCA analysis of the number of TAP families in red algae. Data for 62 TAP families from ten red algae of the taxonomic groups Cyanidiales, Porphyridiophyceae, Florideophyceae, and Bangiophyceae were used to perform a principal component analysis. The *x*-axis shows the first principal component, explaining 74.2% of the variance. The *y*-axis shows the second component (14.2% of variance). The data points are color-coded according to the corresponding taxonomic group. The names of the species are abbreviated using a five-letter code. Cyanidiales (unicellular): CYAME *= Cyanidioschyzon merolae*, CYACA = *Cyanidium caldarium*, GALSU = *Galdieria sulphuraria*, GALPH = *Galdieria phlegrea*. Bangiophyceae: PYRYE = *Pyropia yezoensis*, PORUM = *Porphyra umbilicalis*. Porphyridiophyceae: PORPU = *Porphyridium purpureum*. Florideophyceae (multicellular): GRACH = *Gracilariopsis chorda*, CALTU = *Calliarthron tuberculosum*, CHOCR = *Chondrus crispus*.

**Figure 4 genes-12-01055-f004:**
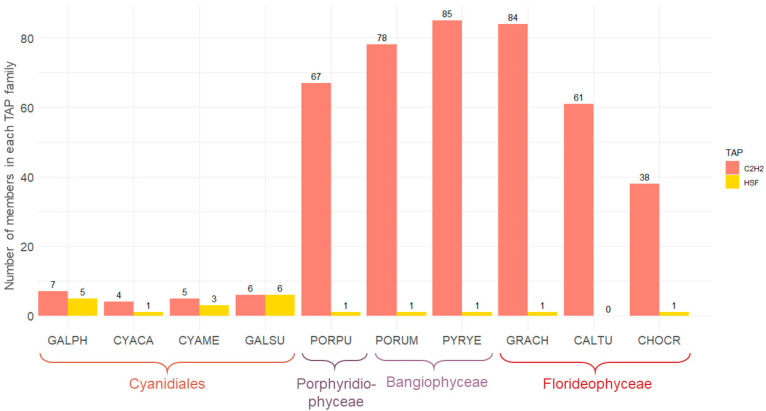
Bar chart illustrating the numbers of C2H2 and HSF family members in red algal species. The numbers of C2H2 and HSF TAP family members are shown for ten red algae species and were generated using TAPscan v3. On the x-axis, the species names are abbreviated in a five-letter code (cf. legend in Figure 3).

**Figure 5 genes-12-01055-f005:**
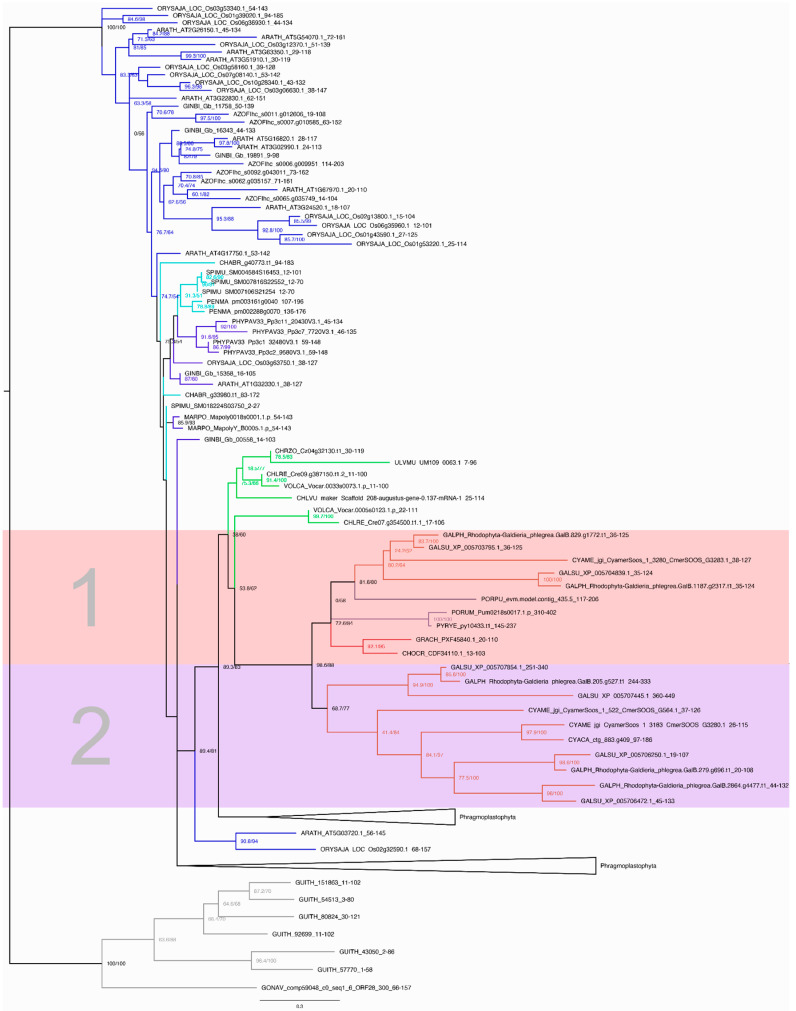
Maximum likelihood tree of the HSF family. The outgroup-rooted tree was calculated using the model LG + R5. The tree contains sequences from 30 species (Appendix A). The names of the species are abbreviated using a five-letter code, where the first three letters indicate the species and the final two the genus (Appendix A). The branches are color-coded according to the taxonomic group: Cyanidiales are shown in light red, Florideophyceae in dark red, Porphyridiophyceae in dark purple, Bangiophyceae in light purple, Chlorophyta in green, streptophyte algae in turquoise, embryophytes in dark blue, and cryptophytes (outgroup) in gray. The subdivided red algae clade is highlighted and numbered. The first subclade (1) contains sequences of the Cyanidiales, Porphyridiophyceae, Bangiophyceae, and Florideophyceae, while the second one (2) contains exclusively sequences of the Cyanidiales. The numbers at the nodes indicate SH-aLRT and ultrafast bootstrap support values. The support values were specified when the corresponding bootstrap value was above 50%. Two groups of the Phragmoplastophyta were collapsed for clarity. The first collapsed Phragmoplastophyta group contains three species of streptophyte algae (one sequence of *Klebsormidium flaccidum*, one of *Penium margaritaceum,* and four *Spirogloea muscicola* sequences). The second collapsed group contains one sequence of *K. flaccidum*.

**Figure 6 genes-12-01055-f006:**
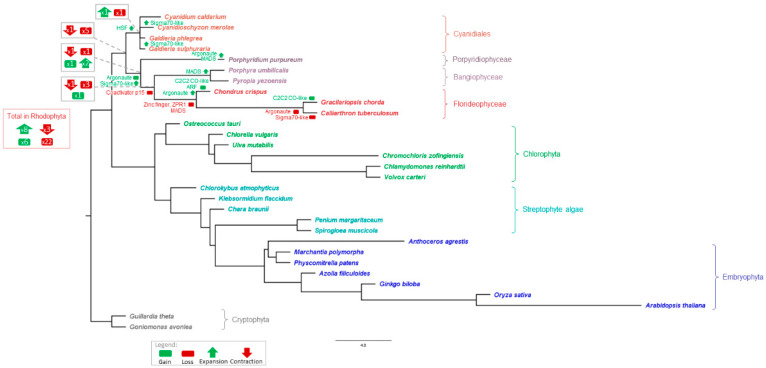
Species tree showing predicted TAP gains, losses, contractions, and expansions. The outgroup-rooted tree is based on the species tree published by the one thousand plant transcriptomes (1kp) consortium [26] and was modified by restriction to the species analyzed here (cf. Materials and Methods, Section 2.5). Selected gains, losses, contractions, and expansions detected by asymmetric Wagner parsimony and discussed in the text are marked by green (gain) and red (loss) boxes, as well as green (expansion) and red (contraction) arrows in the tree. The gray framed boxes at the nodes defining the Cyanidiales, red algae, Bangiophyceae/Florideophyceae/Porphyridiophyceae, and the Bangiophyceae/Florideophyceae respectively, show the total number of gains/losses/expansions/contractions detected at these nodes. The red framed box shows the total number of gains/losses/expansions/contractions within all nodes of the red algae. For complete results, see Appendix A. The species are colored according to the color scheme described in Figure 5, and the brackets indicate taxonomic groups.

## Data Availability

The data presented in this study are available in the Appendix A.

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
