# Peer review of "Signatures of Transcription Factor Evolution and the Secondary Gain of Red Algae Complexity"

_genes, 2021, doi:10.3390/genes12071055_

Round 1
Reviewer 1 Report
This manuscript is an interesting comparison between algal lineages regarding TAP; additionally, the authors’ retraining of TAPseq to better identify TAPs in the algae is a needed addition for the community. It will be no doubt of interest to the readership of Genes, those interested in the evolution of multicellularity, and red algal phycologists for that matter. I have mostly minor comments, which are below.
L53-54 The absence of fossils does not provide proof that the equally old green algae may not have evolved multicellularity as well at an early stage, correct (e.g., https://doi.org/10.1038/s41559-020-1122-9)? Also, the Butterfield citation here is not correct, please correct #8 and check all other references again.
L170-171 Define ZCC and KCM. The readership of Genes is quite broad. I imagine there are even many phycologists who are not savvy regarding these phylogenetic groupings terminology. Define similar terns throughout (e.g., SAR and elsewhere). This is particularly important as there is a high reliance on acronyms through the manuscript.
L213-4 What is the rationale for 25% as opposed to another cut-off value here?
L266 Just a comment: I find it surprising given the skill level at which the authors operate with respect to computational biology and data wrangling that Excel would be used here.
L273 M&M2.5. I am confused here. Am I to understand that the authors did a number of manipulations to an existing tree to essentially make a cartoon to paste the gains and losses illustrated in Figure 5? If so, shouldn't you refer to the “Species tree” as a cartoon? Otherwise, why did the authors’ simply make their own tree? Assuming all the genomes are available, one can identify all gene families shared between the taxa where every taxa is represented just once, then create a tree based on only those single member family constituents.
L364-6 The authors mention “various” yet only list 2 citations. Do you mean “[e.g., 7,19]” or perhaps “[7,19 and references therein]”, or something other?
L375-6 Given that there is only a single single-celled alga in the non-extremophile reds, Porphyridium, that also exhibits this trend this statement should be removed or modified. I understand you deal with this in the Conclusion, but it is confusing here as written. Also, somewhere I think it should be better emphasized that having only a single single-celled red is somewhat confusing with respect to the findings here and that further data is needed in the future to tease out what is going on here with these interesting findings.
L465-6 Is this the most parsimonious assumption that Argonuate evolved independently in Chodrus & Porphyridium/Porphyra rather than lost in Pyropia and the Gracilariales? If this isn’t definitive, it might be worth looking at any other partial or full public genomic data for otherreds and see if anything pops up.
L492 Figure 5. See my comments for L273
Author Response
This manuscript is an interesting comparison between algal lineages regarding TAP; additionally, the authors’ retraining of TAPseq to better identify TAPs in the algae is a needed addition for the community. It will be no doubt of interest to the readership of Genes, those interested in the evolution of multicellularity, and red algal phycologists for that matter. I have mostly minor comments, which are below.
RESPONSE: We thank this reviewer for their kind words.
L53-54 The absence of fossils does not provide proof that the equally old green algae may not have evolved multicellularity as well at an early stage, correct (e.g., https://doi.org/10.1038/s41559-020-1122-9)? Also, the Butterfield citation here is not correct, please correct #8 and check all other references again.
RESPONSE: Thank you for pointing this out, you are right. We changed this section accordingly and checked all references.
L170-171 Define ZCC and KCM. The readership of Genes is quite broad. I imagine there are even many phycologists who are not savvy regarding these phylogenetic groupings terminology. Define similar terns throughout (e.g., SAR and elsewhere). This is particularly important as there is a high reliance on acronyms through the manuscript.
RESPONSE: Indeed, an important point! We have added definitions for ZCC and KCM (L186-189) and for SAR (L220).
L213-4 What is the rationale for 25% as opposed to another cut-off value here?
RESPONSE: We decided for 25% based on previous experience (Wilhelmsson et al., 2017). This value ensures that the existing seed alignment is not overly diluted with novel sequences yet ensures that enough information is included to tweak the HMM to be able to detect non-model organism sequences. We added this argumentation to the revised manuscript.
L266 Just a comment: I find it surprising given the skill level at which the authors operate with respect to computational biology and data wrangling that Excel would be used here.
RESPONSE: Thank you. Sometimes everyday tools still have their use.
L273 M&M2.5. I am confused here. Am I to understand that the authors did a number of manipulations to an existing tree to essentially make a cartoon to paste the gains and losses illustrated in Figure 5? If so, shouldn't you refer to the “Species tree” as a cartoon? Otherwise, why did the authors’ simply make their own tree? Assuming all the genomes are available, one can identify all gene families shared between the taxa where every taxa is represented just once, then create a tree based on only those single member family constituents.
RESPONSE: For the application of the asymmetric Wagner Parsimony a species tree is needed, which contains all species for which gains, losses, expansions and contractions are to be calculated. The results of this analysis were then illustrated in Figure 6. Hence, the tree used for the count analysis is a real tree, not a cartoon tree. The advantage of modifying an existing tree is the availability of reliable branch lengths (in that case from Leebens-Mack et al., 2019). An artificial (constructed) tree would not have branch lengths.
L364-6 The authors mention “various” yet only list 2 citations. Do you mean “[e.g., 7,19]” or perhaps “[7,19 and references therein]”, or something other?
RESPONSE: Changed to [e.g., 7,19].
L375-6 Given that there is only a single single-celled alga in the non-extremophile reds, Porphyridium, that also exhibits this trend this statement should be removed or modified. I understand you deal with this in the Conclusion, but it is confusing here as written. Also, somewhere I think it should be better emphasized that having only a single single-celled red is somewhat confusing with respect to the findings here and that further data is needed in the future to tease out what is going on here with these interesting findings.
RESPONSE: Thank you for bringing this up, this is indeed an important point. We adjusted the respective paragraph accordingly.
L465-6 Is this the most parsimonious assumption that Argonuate evolved independently in Chodrus & Porphyridium/Porphyra rather than lost in Pyropia and the Gracilariales? If this isn’t definitive, it might be worth looking at any other partial or full public genomic data for otherreds and see if anything pops up.
RESPONSE: We rely for this statement on the results of the asymmetric Wagner Parsimony. This kind of analysis is influenced by sampling and hence the most or equal parsimonious explanation might not always be correct. We have added a statement that explains this caveat.
L492 Figure 5. See my comments for L273
RESPONSE: See our response above.
Reviewer 2 Report
Major comments:
- Porphyridium purpureum is classified in the class Porphyridiophyceae, one of member of the subphylum Protorhodophytina with Compsopogonophyceae, Rhodellophyceae and Stylonematophyceae. It is distantly related to the Bangiophyceae and Floridiophyceae (Eurhodophytina). This phylogenetic relationship is supported by many recent molecular studies. In this study, however, the authors used a wrong analytical scheme for Porphyridium as a member of unicellular Bangiophyceae. This wrong phylogenetic scheme brings misleading all interpretations that the authors suggested. Therefore, the entire manuscript have to rewrite based on correct phylogenetic and evolutionary scheme. Because only Cyanidiophyceae lack C2H2, it might be unique gene loss in this taxa. Becasue the Cyanidiophytceae adapted to extreme environments, the loss of C2H2 might be a result of adaptation. As presented results, it is difficult to believe if C2H2 is associated with the morphological complexity in red algae. Same comments for the analysis of HSFs. Further discussion with full reference work is needed.
- The references cited in the manuscript (i.e., 19, 50, 51) did not directly support the biological function of the C2H2 in morphological complexity. Here is one of the referable studies to support the functional relationship between C2H2 and morphological complexity, although it dealt with a land plant; Chen J et al. Control of dissected leaf morphology by a Cys(2)His(2) zinc finger transcription factor in the model legume Medicago truncatula. Proc Natl Acad Sci U S A. 2010 Jun 8;107(23):10754-9. doi: 10.1073/pnas.1003954107. This kind of functional evidence about C2H2 and biological morphology should be discussed, not only the number of TAPs.
- Some members of Compsopogonophyceae and Stylonematophyceae are filamentous and peudofilamentous, therefore, it would be better to add these taxa if the authors wish to suggest the correlation between the multicellularity and TAPs.
- Because the manuscript is focused on the transcription-associated proteins, endosymbiotic plastid evolution should be minimized or deleted.
- Materials and Methods: The TAP classification process in materials and methods was well proceeded to find the red-algal TAPs. However, it seems to be wordy and complicated to follow up the entire process with only texts. Therefore, I recommend representing this process (i.e., 2.1-2.2) using a flowchart (or any schematic workflow) to follow up this process more easy.
- The results of all the analyses should be mentioned in the Results and Discussion. For example, the authors presented NJ tree in supplementary material without any comments in the main text.
- Results and Discussion: There are some duplicated descriptions that should be in the Materials and Methods. For example, in Results 3.1, the first two sentences were already mentioned in Materials and Methods.
- Results 3.4: There is no result and discussion for the C2H2 phylogeny, but only in Conclusion. It should be added in the Results and Discussion.
Minor comments:
Lines 170-171, and Line 400-401: There are no descriptions of abbreviations, ZCC grade, and KMC grade. Please add a short description or full name of the abbreviations because it might be unfamiliar to the readers who don’t have a background about the taxonomy of the Chloroplastida.
Line 174: What is the subject of ‘improving the sensitivity selected profile HMMs’? all of genes? TAPs? This subtitle is too obscure to infer the corresponding process. Please change this subtitle to a more specified one.
Line 265: Change ’Florideophyceae, Bangiophyceae and Cyanidiales’ to ’red algal’.
Line 289: Change ’calculated’ to ’constructed’.
Lines 289 and 314: Delete ’cf.’
Lines 289-290: The figures including supplementary materials should be cited in the main text.
Lines 300-302: Please indicate the model test and selection for ML analysis.
Line 310: Cite supplementary Tables or Figures.
Line 316, ’these six profile HMMs’: Does ’these’ refers the above five profiles (AP2, bZIP, bHLH, C2H2, and HMG)?
Line 321-323: “~ sensitivity for all of the five domains had increased (Table S1 and S2) ”. How much the sensitivity was increased? Please state the number not only the citation of Tables.
Lines 336-337: Change ’members’ to ’species’.
Line 341 and 347: Add references.
Line 350: The title of Figure2 is quite ambiguous to understand what are the raw data of PCA. Please specify the data source of PCA in the title, for example, ‘PCA analysis of the number of TAP families in red algae’.
Line 375: There is no reference in the sentence “This trend was also observed for red algae in the current study, ~”. The reference which corresponds to ‘the current study’ must be cited here.
Line 404: Change ’TAP family HSF’ to ’HSF family’.
Line 405: Delete ’(used as outgroup)’.
Line 406: Add the support values.
Line 412-413: I could not understand the second scenario. Cyanidiales is the first derived monophyletic lineage in red algae.
Line 414: Delete ’family’.
Line 414: ture? Are you sure?
Lines 417-433: Please merge the second and third paragraphs.
Line 424: Delete ’i.e.’.
Lines 420-426: The authors detected 8 HSFs in S. muscicola and 2 in C. braunii. However, in Fig. 4, there are 4 HPFs of S. muscicola 1 of C. braunii. Please check the numbers in the text and figure. Also change ’many’ (line 425) to a specific number.
Line 426-427: The HSF pattern (high number) of S. muscicola indicate the expansion due to gene duplication events. But, in lines 431-433, it is due to adaptations to environmental conditions.
Lines 429-430: Please cite references in the case of more TAPs present more morphologically complex organisms.
Line 431: Change ’charophyte’ to ’streptophyte’.
Lines 437-438: Rephrase the sentence.
Lines 478-480: Please check the number of this sentence, 6.801 and 1.448. It should be changed to 6,801 and 1,448.
Lines 507-509: Please check the number of this sentence, ”~ ca. 1.900 Ma ago.”. And, add the reference(s).
Line 510: Delete ’(Table S4)’.
Citation of Figures, Tables and Supplementary materials in the maintext:
- All tables and figures should be cited in numerical order. Rearrange the figures and tables. For example, cite figure 1 then 2, 3 etc. Same in Supplementary materials. Table S3 need to be Table S1.
- Add ’Supplementary’ in the text when cite the supplementary materials.
- Some of the figures and supplementary materials are cited in the Materials and Methods. It should be in Results and Discussion.
- Figure 1: Did you included Mesostigma virida in your analysis? If not, delete from the figure.
- Figure 4: This figure seems not ready for publication. Revise it with full species names. Check the substitution model. LG+G+I (Materials and Methods) or LG+R5 (figure legend). Also revise the figure legend. The node support vales are ML bootstrap values and Bayesian posterior probabilities. The support values for each node are difficult to recognize. The decimal point is not needed. Bootstrap values < 50% and Bayesian posterior probabilities < 0.90 are not statistically significant, so it can be deleted.
Author Response
Reviewer 2
- Porphyridium purpureum is classified in the class Porphyridiophyceae, one of member of the subphylum Protorhodophytina with Compsopogonophyceae, Rhodellophyceae and Stylonematophyceae. It is distantly related to the Bangiophyceae and Floridiophyceae (Eurhodophytina). This phylogenetic relationship is supported by many recent molecular studies. In this study, however, the authors used a wrong analytical scheme for Porphyridium as a member of unicellular Bangiophyceae. This wrong phylogenetic scheme brings misleading all interpretations that the authors suggested. Therefore, the entire manuscript have to rewrite based on correct phylogenetic and evolutionary scheme. Because only Cyanidiophyceae lack C2H2, it might be unique gene loss in this taxa. Becasue the Cyanidiophytceae adapted to extreme environments, the loss of C2H2 might be a result of adaptation. As presented results, it is difficult to believe if C2H2 is associated with the morphological complexity in red algae. Same comments for the analysis of HSFs. Further discussion with full reference work is needed.
RESPONSE: Indeed, and thank you for pointing this out! We adjusted the manuscript by assigning Porphyridium purpureum to the Porphyridiophyceae. All corresponding parts of the text have been changed, as well as the figures and tables.
- The references cited in the manuscript (i.e., 19, 50, 51) did not directly support the biological function of the C2H2 in morphological complexity. Here is one of the referable studies to support the functional relationship between C2H2 and morphological complexity, although it dealt with a land plant; Chen J et al. Control of dissected leaf morphology by a Cys(2)His(2) zinc finger transcription factor in the model legume Medicago truncatula. Proc Natl Acad Sci U S A. 2010 Jun 8;107(23):10754-9. doi: 10.1073/pnas.1003954107. This kind of functional evidence about C2H2 and biological morphology should be discussed, not only the number of TAPs.
RESPONSE: Thank you; we added this study in the section about coincidence of C2H2 numbers and morphological complexity.
- Some members of Compsopogonophyceae and Stylonematophyceae are filamentous and peudofilamentous, therefore, it would be better to add these taxa if the authors wish to suggest the correlation between the multicellularity and TAPs.
RESPONSE: Of course, you are right, we look forward to investigate further red algal species in subsequent studies regarding the presence of TAPs and their morphology.
- Because the manuscript is focused on the transcription-associated proteins, endosymbiotic plastid evolution should be minimized or deleted.
RESPONSE: We have minimized the corresponding section in the introduction accordingly (L46-56).
- Materials and Methods: The TAP classification process in materials and methods was well proceeded to find the red-algal TAPs. However, it seems to be wordy and complicated to follow up the entire process with only texts. Therefore, I recommend representing this process (i.e., 2.1-2.2) using a flowchart (or any schematic workflow) to follow up this process more easy.
RESPONSE: Thank you for the suggestion, a flowchart simplifies the understanding of the workflow. We have added it as figure 2 to section 2.2.
- The results of all the analyses should be mentioned in the Results and Discussion. For example, the authors presented NJ tree in supplementary material without any comments in the main text.
RESPONSE: We have now mentioned all the phylogenetic trees in the main text (L444 & L479). Furthermore, we added the reference for the supplementary file S1 into section 3.1 (L358).
- Results and Discussion: There are some duplicated descriptions that should be in the Materials and Methods. For example, in Results 3.1, the first two sentences were already mentioned in Materials and Methods.
RESPONSE: We deleted the first paragraph of section 3.1 to avoid duplications.
- Results 3.4: There is no result and discussion for the C2H2 phylogeny, but only in Conclusion. It should be added in the Results and Discussion.
RESPONSE: The description of the C2H2 phylogeny is in section 3.4 L478-484.
Minor comments:
Lines 170-171, and Line 400-401: There are no descriptions of abbreviations, ZCC grade, and KMC grade. Please add a short description or full name of the abbreviations because it might be unfamiliar to the readers who don’t have a background about the taxonomy of the Chloroplastida.
RESPONSE: We have added definitions for ZCC and KCM (L186-189).
Line 174: What is the subject of ‘improving the sensitivity selected profile HMMs’? all of genes? TAPs? This subtitle is too obscure to infer the corresponding process. Please change this subtitle to a more specified one.
RESPONSE: We changed it to: “Improving the sensitivity of selected profile HMMs for algae by adding diverse algal sequences”.
Line 265: Change ’Florideophyceae, Bangiophyceae and Cyanidiales’ to ’red algal’.
RESPONSE: Done.
Line 289: Change ’calculated’ to ’constructed’.
RESPONSE: Changed to inferred.
Lines 289 and 314: Delete ’cf.’
RESPONSE: Done.
Lines 289-290: The figures including supplementary materials should be cited in the main text.
RESPONSE: We now cite all phylogenetic trees in the main text (L444 & L479).
Lines 300-302: Please indicate the model test and selection for ML analysis.
RESPONSE: Added in L328.
Line 310: Cite supplementary Tables or Figures.
RESPONSE: We have deleted this section (see 7.).
Line 316, ’these six profile HMMs’: Does ’these’ refers the above five profiles (AP2, bZIP, bHLH, C2H2, and HMG)?
RESPONSE: Yes, “these” refers to the five TAP families. Since the TAP family bZIP has two corresponding profile HMMs, there is a total of six profile HMMs for these five TAP families. We have adjusted the sentence to make this difference clearer.
Line 321-323: “~ sensitivity for all of the five domains had increased (Table S1 and S2)”. How much the sensitivity was increased? Please state the number not only the citation of Tables.
RESPONSE: We added: “…, with a minimum of two and a maximum of twelve additional true positives detected “. (L350-351)
Lines 336-337: Change ’members’ to ’species’.
RESPONSE: Done.
Line 341 and 347: Add references.
RESPONSE: In Line 374 we added the reference Collén et al., 2013. By adjusting the classification of Porphyridium, the second reference became obsolete.
Line 350: The title of Figure2 is quite ambiguous to understand what are the raw data of PCA. Please specify the data source of PCA in the title, for example, ‘PCA analysis of the number of TAP families in red algae’.
RESPONSE: Changed to the suggested figure title.
Line 375: There is no reference in the sentence “This trend was also observed for red algae in the current study, ~”. The reference which corresponds to ‘the current study’ must be cited here.
RESPONSE: This sentence has been rephrased, including a change to “this study” to make clear that we refer to the data presented in the present manuscript.
Line 404: Change ’TAP family HSF’ to ’HSF family’.
RESPONSE: Done.
Line 405: Delete ’(used as outgroup)’.
RESPONSE: Done.
Line 406: Add the support values.
RESPONSE: We added that both, the SH-aLRT and ultrafast bootstrap support values are 100%.
Line 412-413: I could not understand the second scenario. Cyanidiales is the first derived monophyletic lineage in red algae.
RESPONSE: In scenario one losses of HSF members occurred in Bangiophyceae, Porphyridiophyceae and Florideophyceae, while in scenario two expansions occurred in the Cyanidiales after the split from the other red algal lineages.
RESPONSE: Done.
Line 414: ture? Are you sure?
RESPONSE: We rephrased this sentence.
Lines 417-433: Please merge the second and third paragraphs.
RESPONSE: Done.
Line 424: Delete ’i.e.’.
RESPONSE: Done.
Lines 420-426: The authors detected 8 HSFs in S. muscicola and 2 in C. braunii. However, in Fig. 4, there are 4 HPFs of S. muscicola 1 of C. braunii. Please check the numbers in the text and figure. Also change ’many’ (line 425) to a specific number.
RESPONSE: In Figure 5 we have collapsed two groups of the Phragmoplastophyta for clarity. The first collapsed group includes one sequence of K. flaccidum, one of P. margaritaceum, and four S. muscicola sequences. The second collapsed group includes one sequence of K. flaccidum. This information is presented in the legend to Figure 5 (L495-498). We exchanged the word “many” with “three S. muscicola sequences, and four sequences in the collapsed groups “.
Line 426-427: The HSF pattern (high number) of S. muscicola indicate the expansion due to gene duplication events. But, in lines 431-433, it is due to adaptations to environmental conditions.
RESPONSE: Lineage-specific expansions can be characterized by gene duplication events as adaptions to environmental conditions. To make this clear we have modified the sentence in L470: “This indicates lineage-specific expansion characterized by gene duplication events, as in the Cyanidiales “.
Lines 429-430: Please cite references in the case of more TAPs present more morphologically complex organisms.
RESPONSE: Added Lang et al., 2010 as reference.
Line 431: Change ’charophyte’ to ’streptophyte’.
RESPONSE: Done.
Lines 437-438: Rephrase the sentence.
RESPONSE: Done.
Lines 478-480: Please check the number of this sentence, 6.801 and 1.448. It should be changed to 6,801 and 1,448.
RESPONSE: Done.
Lines 507-509: Please check the number of this sentence, ”~ ca. 1.900 Ma ago.”. And, add the reference(s).
RESPONSE: We changed the number; the estimate is based on the cited work, Strassert et al., 2021.
Line 510: Delete ’(Table S4)’.
RESPONSE: Done.
Citation of Figures, Tables and Supplementary materials in the maintext:
- All tables and figures should be cited in numerical order. Rearrange the figures and tables. For example, cite figure 1 then 2, 3 etc. Same in Supplementary materials. Table S3 need to be Table S1.
RESPONSE: Thank you for catching that, we have adjusted it.
- Add ’Supplementary’ in the text when cite the supplementary materials.
RESPONSE: Done.
- Some of the figures and supplementary materials are cited in the Materials and Methods. It should be in Results and Discussion.
RESPONSE: We have now mentioned all the phylogenetic trees in the main text (L444 & L479). Furthermore, we added the reference for the supplementary file S1 in section 3.1 (L358).
- Figure 1: Did you included Mesostigma virida in your analysis? If not, delete from the figure.
RESPONSE: That was by mistake. We have adjusted the figure accordingly.
- Figure 4: This figure seems not ready for publication. Revise it with full species names. Check the substitution model. LG+G+I (Materials and Methods) or LG+R5 (figure legend). Also revise the figure legend. The node support vales are ML bootstrap values and Bayesian posterior probabilities. The support values for each node are difficult to recognize. The decimal point is not needed. Bootstrap values < 50% and Bayesian posterior probabilities < 0.90 are not statistically significant, so it can be deleted.
RESPONSE: We have revised figure 5 for readability, and adjusted labels and support values. Furthermore, we deleted all support values where the bootstrap value was < 50%. The model LG+R5 was used to calculate the shown ML tree, LG was determined by ProtTest and was used to calculate the BI tree. These models are mentioned in Material and Methods (section 2.6), as well as in the figure legends.
Round 2
Reviewer 2 Report
The authors addressed some comments that I raised, however, it is still not satisfactory, because it is reasonable to interpret the expansion/reduction of TAP family (i.e., TAP families, C2H2, HSF) are Cyanidiales-specific feature rather than correlation between these with unicellularity vs multicellularity. Indeed, the authors agreed this in lines 439-442 as “This indicates lineage-specific expansion characterized by gene duplication events, as in the Cyanidiales. In general, it is observed that the distribution of the HSF family in red algae is not consistent with the trend that more TAP members are present in morphologically more complex organisms [16], since unicellular red algae have more HSF members than multicellular species.”
However, the authors followed contradictory conclusion as “Based on the tree, it can be concluded that expansions of C2H2 have occurred in multicellular Bangiophyceae and Florideophyceae after separation from the Cyanidiales lineage. In comparison to unicellular species, these expansions might be related to the increased morphological complexity in the respective groups.” in line 447-451. It seems to me that the authors interpret the results with prejudice of unicellular vs multicellular issue. This is a serious flow that needs to be corrected through the entire manuscript, because unicellular Porphyridium usually closer to the multicellular Bangiales than the unicellular Cyanidiales in their results (Figs 3-6). Along this line, the title should be modified with more precise representation.
Figure 1 & 6: The back-bone trees in these figures are still not followed the widely accepted red algal phylogenetic relationship because they came from green plants targeted phylogenetic study (1KP project). Particularly, Porphyridiophyceae and Bangiophyceae are not monophyletic group, and these are distant in different Subphylum (e.g., doi: 10.1016/j.cub.2017.04.054). Bangiophyceae and Florideophyceae are a strong monophyletic group. These back-bone trees should be revised. Based on revised species tree, Fig 6 and its interpretation have to be revised accordingly.
Figure 3: Change ’Multicellular Bangiophyceae’ to ’Bangiophyceae’ in the legend on the figure.
Author Response
The authors addressed some comments that I raised, however, it is still not satisfactory, because it is reasonable to interpret the expansion/reduction of TAP family (i.e., TAP families, C2H2, HSF) are Cyanidiales-specific feature rather than correlation between these with unicellularity vs multicellularity. Indeed, the authors agreed this in lines 439-442 as “This indicates lineage-specific expansion characterized by gene duplication events, as in the Cyanidiales. In general, it is observed that the distribution of the HSF family in red algae is not consistent with the trend that more TAP members are present in morphologically more complex organisms [16], since unicellular red algae have more HSF members than multicellular species.”
However, the authors followed contradictory conclusion as “Based on the tree, it can be concluded that expansions of C2H2 have occurred in multicellular Bangiophyceae and Florideophyceae after separation from the Cyanidiales lineage. In comparison to unicellular species, these expansions might be related to the increased morphological complexity in the respective groups.” in line 447-451. It seems to me that the authors interpret the results with prejudice of unicellular vs multicellular issue. This is a serious flow that needs to be corrected through the entire manuscript, because unicellular Porphyridium usually closer to the multicellular Bangiales than the unicellular Cyanidiales in their results (Figs 3-6). Along this line, the title should be modified with more precise representation.
RESPONSE: We agree with the reviewer that different evolutionary scenarios are possible. Hence, we have expanded the discussion of the C2H2 family to point out that loss (in the Cyanidiales) is an option. Nevertheless, we also present arguments to make clear that we consider the alternative hypothesis, independent gain after the divergence from Cyanidiales, more probable. We also comment on the possibility that the abundance of gene members in the MRCA of Porphyridiophyceae/Bangiophyceae/Florideophyceae might have laid the ground for multicellularity to arise.
We would also like to point out that this paper is to appear in a special issue that deals with the uni-/multicellular transition and therefore that it is important that we discuss TAP evolution in the context of this evolutionary question. We agree that coincidence of TF abundance and morphological complexity is not a proof that one is causative for the other. Yet, this hypothesis has been described in literature several times and in our present study we cite this work and add our own observation. In addition to the altered discussion outlined above, we also changed the title to be less suggestive, and added the caveat that more genomes of unicellular Porphyridiophyceae and Bangiophyceae will be needed to draw conclusions.
Figure 1 & 6: The back-bone trees in these figures are still not followed the widely accepted red algal phylogenetic relationship because they came from green plants targeted phylogenetic study (1KP project). Particularly, Porphyridiophyceae and Bangiophyceae are not monophyletic group, and these are distant in different Subphylum (e.g., doi: 10.1016/j.cub.2017.04.054). Bangiophyceae and Florideophyceae are a strong monophyletic group. These back-bone trees should be revised. Based on revised species tree, Fig 6 and its interpretation have to be revised accordingly.
RESPONSE: In the revised version, we adjusted the trees that are the basis for both, Figure 1 and Figure 6, representing the Porphyridiophyceae as sister lineage to the Bangiophyceae/Florideophyceae clade. With this new species tree, we recalculated the asymmetric Wagner Parsimony and adjusted the results accordingly.
Figure 3: Change ’Multicellular Bangiophyceae’ to ’Bangiophyceae’ in the legend on the figure.
RESPONSE: Done.